# LEARNING TO REMEMBER FROM A MULTI-TASK TEACHER

## ABSTRACT

Recent studies on catastrophic forgetting during sequential learning typically focus on fixing the accuracy of the predictions for a previously learned task. In this paper we argue that the outputs of neural networks are subject to rapid changes when learning a new data distribution, and networks that appear to "forget" everything still contain useful representation towards previous tasks. Instead of enforcing the output accuracy to stay the same, we propose to reduce the effect of catastrophic forgetting on the representation level, as the output layer can be quickly recovered later with a small number of examples. Towards this goal, we propose an experimental setup that measures the amount of representational forgetting, and develop a novel meta-learning algorithm to overcome this issue. The proposed meta-learner produces weight updates of a sequential learning network, mimicking a multi-task teacher network's representation. We show that our meta-learner can improve its learned representations on new tasks, while maintaining a good representation for old tasks.

## 1 INTRODUCTION

An intelligent agent needs to deal with a dynamic world and is typically presented with sequential tasks that are highly correlated in time yet constantly changing. Newborns learn to build generic representations from video and audio streaming input. Kids can learn highly skilled tasks such as skiing and swimming sequentially without worrying about forgetting one another. Humans seem to have a robust way of learning representations from sequential inputs (and tasks), yet state-of-the-art machine learning algorithms rely heavily on uniformly sampled training examples from the same distribution.

One of the major challenges in sequential learning of neural networks is the issue of *catastrophic forgetting* (McCloskey & Cohen, 1989)–after a neural network is trained on a new task, its performance on old tasks drops significantly. Despite several attempts, this problem remains unsolved. Explicit weight regularization methods (Evgeniou & Pontil, 2004; Kirkpatrick et al., 2016; Lee et al., 2017) often rely on simplistic assumptions on the shape of the weight posterior distribution. Model compression methods (Serrà et al., 2018; Fernando et al., 2017; Mallya & Lazebnik, 2018) seem promising on existing benchmarks, however the underlying mechanism is to train small individual networks that may lack global cooperation, a limiting factor when learning a large number of classes towards a generic representation. Generative models (Shin et al., 2017; Kemker & Kanan, 2018; Venkatesan et al., 2017) seem to be a natural choice; however, training a high quality generative model is far from trivial, oftentimes more complex than training the original network itself.

Despite the variety of models that have been proposed, there seems to be a lack of general understanding on what kind of knowledge is being forgotten and to what extent it can be recovered. Recent research places much of its focus on maintaining the output performance of previous tasks. In this paper we argue that this can be misleading since the output layer of a network is very sensitive to changes in the output distribution. Instead, here we would like to understand how much of the performance drop is related to the lack of training on previous output layers versus the loss of information in the newly learned representation. Towards this goal, we exploit a linear decoding layer to measure the amount of catastrophic forgetting on the representation level. This gives us insights on whether the drop in performance is likely to be recovered by re-learning the output layer from very few examples.

Motivated by recent progress on meta-learning (Jaderberg et al., 2017; Andrychowicz et al., 2016; Metz et al., 2018), in this paper we propose to learn a weight update rule to overcome representational

forgetting. our meta-learning algorithm learn such a rule by rolling out many sequential learning experiences during training. In human language acquisition, it is found that children who lost their first language maintain similar brain activation to bilingual speakers (Pierce et al., 2014). Inspired by this fact, we propose a novel meta-learning algorithm that tries to mimic a multi-task teacher network's representation, an offline oracle in our sequential learning setup, since multi-task learning has simultaneous access to all tasks whereas our sequential learning algorithm only has access to one task at a time.

In summary, the contributions of our paper are two-fold. First, we propose a measure of catastrophic forgetting at the representation level, which provides more insight on the amount of previous knowledge forgotten in the new task. Second, we develop a new meta-learning algorithm that can predict weight updates that are less prone to catastrophic forgetting than standard backpropagation. We demonstrate the effectiveness of our approach on the MNIST (Lecun et al., 1998), FashionM-NIST (Xiao et al., 2017), CIFAR-10, and CIFAR-100 (Krizhevsky, 2009) datasets, and find that our meta-learner is able to generalize to unseen object classes from meta-training.

The rest of the paper is organized as follows: we first survey existing literature in Section 2; Section 3 describes representational forgetting in deep neural networks, and our experimental setup to measure it; Section 4 details our proposed meta-learning algorithm, followed by experimental results in Section 5.

## 2 RELATED WORK

Catastrophic forgetting in sequential learning has been studied in the early literature of neural networks (McCloskey & Cohen, 1989; Ratcliff, 1990; French, 1991; Mcrae & Hetherington, 1993; French, 1999). The degree of forgetting is usually measured in terms of the amount of time, typically the number of iterations, saved to relearn the old task. This metric has several drawbacks as the number of iterations can be fairly sensitive to the choice of optimization hyper-parameters and network architecture. Furthermore, the network may never recover fully the original performance.

With the recent success of deep learning, the issue of catastrophic forgetting has regained attention in the research community. Unlike classical methods, recent papers measure the old task performance immediately after training on the new task (Kirkpatrick et al., 2016; Goodfellow et al., 2013; Li & Hoiem, 2018). In this setting, networks that are trained in a sequential manner over tasks suffer dramatically, due to the fact that the previous output classifier branches are no longer tuned to the newly learned representations.

In this paper we follow the early literature, and allow the network to relearn on old tasks, as we believe it is natural to review the old task before testing on it again. But instead of counting the number of iterations to recover the old task performance, we propose to train on top of the newly learned representation using a small decoding model to perform the old task. We believe this measure well captures the amount of representational forgetting. To this end, we exploit a simple linear readout layer, as it is relatively fast to train and is more robust than measuring the number of recovering steps.

One way to address catastrophic forgetting is through explicit regularization. Evgeniou & Pontil (2004) add an L2 regularizer to ensure that the new weights do not "drift" away from the old weights. Elastic weight consolidation (Kirkpatrick et al., 2016) computes the strength of the regularizer on each weight dimension using a diagonal approximation of the Fisher information matrix. Zenke et al. (2017) propose to directly approximate the regularization strength online. Lee et al. (2017) incrementally match the moments of posterior Gaussian distributions of new tasks. These methods are often motivated from a simple quadratic loss surface, and can be potentially limiting their ability to deal with more complex learning dynamics. Regularization can also be imposed on the activation level: Learning without Forgetting (Li & Hoiem, 2018) regularizes the network such that the logits of the new examples on the old classifier remain similar. While this framework is more flexible, using the old activations to distill new tasks can be less informative if the network has not seen enough classes.

In contrast to continuous regularization, model compression based approaches (Mallya & Lazebnik, 2018; Fernando et al., 2017; Rusu et al., 2016; Yoon et al., 2018) discretely allocate certain capacity of a network towards learning new tasks. PackNet (Mallya & Lazebnik, 2018) applies network pruning in between sequential tasks, so that the pruned neurons can be re-allocated. PathNet (Fernando et al., 2017) uses genetic algorithms to select pathways of the network for reuse. HAT (Serrà et al., 2018)

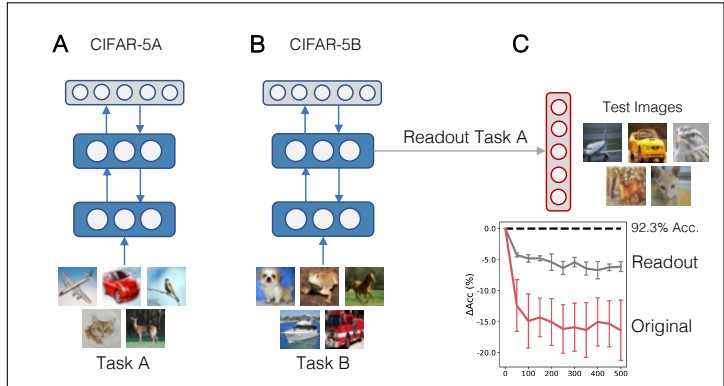

Figure 1: Using a readout layer to measure catastrophic forgetting on representations. **A)** A network is first pre-trained on Task A. **B)** Then it is finetuned on Task B. **C)** We then feed in Task A training data to the network, and record the representation at the last layer prior to the classification layer. We re-train a readout layer to recover Task A output. Test accuracy on Task A is recorded as "Readout", and original classification layer accuracy is recorded as "Original". Chance is 20%.

learns a hard binary mask for each weight connection. Rusu et al. (2016) add connections from old frozen modules towards newly allocated modules, at the price of learning more intermediate layers, thus scaling quadratically with the number of tasks. Yoon et al. (2018) propose to dynamically prune and allocate neurons at the same time. In comparison, our meta-learner can also be interpreted as implicitly learning to perform dynamic capacity re-allocation without increasing the network size; however, our approach does not choose a discrete set of neurons or synapses to update, but try to mimic the activation responses from a multi-task network. This allows the algorithm to learn to implicitly allocate weight subspaces for learning the new tasks.

Another class of methods store a subset of the old data, so that the old task can be jointly trained. iCaRL Rebuffi et al. (2017) propose to choose representative exemplars of old tasks. Gradient episodic memory (Lopez-Paz & Ranzato, 2017) stores old examples and makes sure that the new example only updates in the direction that agrees with the gradient directions of old examples. Sprechmann et al. (2018) explore a learnable memory architecture to dynamically store and retrieve examples facilitating sequential learning. Though effective, storing raw data points costs additional storage and it may also lack biological plausibility. To address the issue of data storage, generative models have also been used to avoid storing old raw data (Shin et al., 2017; Venkatesan et al., 2017; Kemker & Kanan, 2018; van der Ven & Tolias, 2018). Although generative models enjoy the benefits of data storage based models, the final performance heavily depends on the quality of the generated data, because training a competitive generative model itself may be more complex and take more capacity than the original network.

Our proposed model is inspired from prior work in meta-learning: using a learned parameterized weight update rule (Bengio et al., 1990) instead of standard optimization methods. Synthetic gradient (Jaderberg et al., 2017) uses an MLP to predict the gradient direction when performing forward passes, allowing asynchronous weight updates across layers. Andrychowicz et al. (2016); Ravi & Larochelle (2017) use a recurrent network to predict updates. Metz et al. (2018) propose to learn an unsupervised learning rule based solely on activations. Miconi et al. (2018; 2019) combine a learned Hebbian plasticity rule with learned weights. The largest difference between our proposed model and prior work is the fact that instead of predicting the gradients or improve the task performance at the end of the training episode, our meta-learner is trained with a teacher network's activation as supervision.

## 3 REPRESENTATIONAL FORGETTING IN SEQUENTIAL LEARNING

Sequential learning is the process of learning tasks sequentially without revisiting previous tasks. For simplicity, in this paper we study the setting of two tasks, where a network is first presented with *Task A* and then *Task B*. The exposure to Task A can bring positive benefits towards learning Task B if the two tasks are similar. This is a property often studied in the transfer learning literature. Unfortunately, it is challenging to maintain the initial performance of the model learned on Task A (when updating it to perform well on task B), especially when the old task environment is not available to the agent. For example, a commercial robot needs to adapt and learn in new environments while the original training data cannot be shipped together with the learning algorithm.

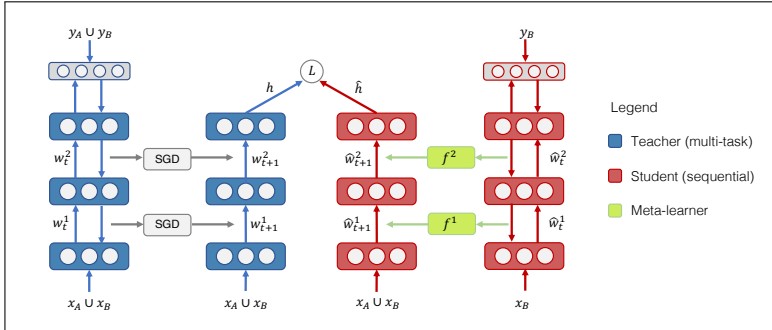

Figure 2: Overview of our purposed method during one training step: 1) the teacher network is updated using SGD on multi-task data; 2) the student network is updated using the meta-learner on Task B data only; 3) the multi-task data are fed into both networks and we record the representations as $h$ and $\hat{h}$; 4) the meta module is then updated to minimize the difference between $h$ and $\hat{h}$. At meta-test time, the teacher network is no longer present, and we update the student network solely with the trained meta-learner for the entire training sequence.

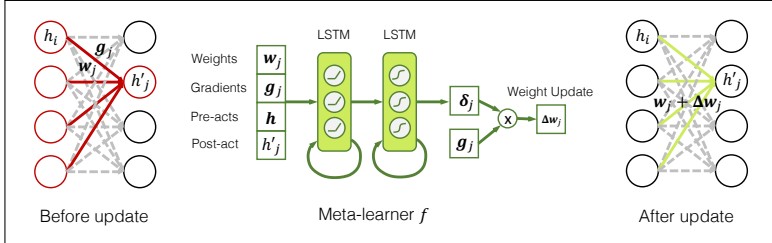

Figure 3: Our meta-learner module is a stacked LSTM network. For a given output neuron $h'_j$, the meta-network takes in its activation, its pre-activations, current weights and gradients, and output the weight updates assiciated with $h'_j$. Weights in the same layer will be shared a single meta-network, while different layers will have different nmeta-networks.

Catastrophic forgetting (McCloskey & Cohen, 1989) occurs when a learning agent forgets the old task after adapting to the new task. While recent literature solely focuses on the output performance of an agent on the previous tasks, it is unclear whether the agent "truly forgets" the prior experiences, or only the output layers are miscalibrated due to learning the new task. The latter issue has an easy solution as simply training the output classifier on the old task for a few iterations can recover most of the performance loss. This is similar to human revisiting previously learned skills.

In this paper we study how much forgetting occurs at the representation level. Towards this goal, we propose the experiment setup illustrated in Figure 1 where a network is first trained on Task A, and it learns the representation of the inputs, referred to as the final layer prior to the classification head. Next, a different task, Task B is introduced. In the second stage of training, the model no longer has access to Task A. It then fine-tunes all its layers on Task B, since learning a better representation will be useful for the new task. To test the amount of forgetting on Task A, we train a linear readout layer on the newly learned representations, using the training examples from Task A, and evaluate the test performance.

We repeatedly do this readout training throughout the learning of task B. In the beginning, the readout performance is close to the original Task A performance, and later, the features become less selective towards Task A. As shown in Figure 1, we indeed observe that the readout accuracy of Task A is constantly decreasing as the learning process goes on. In contrast to measuring the output performance directly ("Readout" vs. "Original" in Figure 1), as was done in prior catastrophic forgetting literature, the level of forgetting here is not as dramatic, suggesting a portion of forgetting happens due to the mis-calibration of the output layer. Representational forgetting is still significant–for the binary classification problem illustrated in Figure 1, readout accuracy on Task A drops over 15% after 300 iterations of training on Task B.

Meta-learning is a general tool for us to learn a new learning algorithm with desired properties. In the next section, we propose a novel meta-learning algorithm that directly addresses the issue of representational forgetting.

## 4 LEARNING FROM A MULTI-TASK TEACHER

As the learning process goes on in Task B, each gradient descent step can potentially erase useful features for the old task. Continuous regularization (Evgeniou & Pontil, 2004; Kirkpatrick et al., 2016; Zenke et al., 2017) or discrete model pruning based methods (Mallya & Lazebnik, 2018; Fernando et al., 2017; Rusu et al., 2016; Yoon et al., 2018), as discussed in Section 2, can hurt the capacity of learning new tasks and limit the sharing of a distributed representation. Manually designing a sophisticated learning objective is not obvious, thus we are interested in learning a *learning rule* that can predict dynamic weight updates alongside the training of Task B.

For simplicity, we describe the setting of a fully connected layer, shown in Figure 3, where $i$ indexes pre-activations and $j$ post-activations. A weight synapse $w_{i,j}$ connects to its pre-activation $h_i$ and post-activation $h'_j$. Let $g_{i,j}$ be the gradient of the connection obtained through regular backpropagation. Our meta-learner $f$ is implemented as a long short-term memory (LSTM) network that takes as inputs the weight connection, gradients, inputs and outputs, similar to what has been done by Metz et al. (2018). The network intuitively predicts a dynamic gating that modulates the plasticity of synapses: $\delta_{\cdot,j} = f(\mathbf{w}_{\cdot,j}, \mathbf{g}_{\cdot,j}, \mathbf{h}_\cdot, h'_j; \theta)$, which is then multiplied with the original gradients to form the updates of the synapse: $\Delta w_{\cdot,j} = \delta_{\cdot,j} \cdot \mathbf{g}_{\cdot,j}$. Finally, the weights are updated in a similar way to SGD, $\hat{\mathbf{w}}_{\cdot,j} = \mathbf{w}_{\cdot,j} - \alpha \Delta \mathbf{w}_{\cdot,j}$, where $\alpha$ is the learning rate. We can generalize this setup to convolutional layers as well by considering all filter locations together, with more details in Section 5.1.

Learning this meta-learner, is however a non-trivial task. In (Metz et al., 2018), the learning process is unrolled just like a recurrent neural network for a large number of steps, and meta-learning is done using backpropagation through time, which is inefficient since each meta-update can only be done in the outer loop.

A key insight of our paper is that, we can actually learn from a multi-task teacher network as if it is an oracle to our sequential learner. Similar to knowledge distillation (Hinton et al., 2015), where a student network learns activation patterns from a teacher network, here, we would like to train the meta-learner such that the student network has a similar activation pattern compared to the teacher network. This avoids using the final accuracy of an episode as the supervision signal, which can be very inefficient.

The idea of using a multi-task network as a teacher is inspired by the human language acquisition literature. It was found that adopted children who were separated from the Chinese language (their birth language) at around one year old on average, still maintain activations in their left superior temporal gyrus similar to French/Chinese bilingual speakers at the age of 9 to 17, even though they have no exposure to Chinese for 13 years on average (Pierce et al., 2014). We draw a parallel in our sequential learning framework here: whereas the network is allowed to forget the old task on the output level, the learned representation should resemble the one learned by a multi-task network.

**Algorithm 1** Learning to Remember from a Multi-Task Teacher

**Require:** $w_0, \mathcal{D}_A, \mathcal{D}_B$
**Ensure:** $\theta$ (Meta-parameters)
1: **for** $i = 1 \dots N$ **do**
2:     // Reinitialize teacher and student networks
3:     $w \leftarrow w_0$
4:     $\hat{w} \leftarrow w_0$
5:     Meta-learner resets hidden state;
6:     // Reinitialize T-BPTT step $s$ to zero
7:     $s \leftarrow 0$
8:     **for** $t = 1 \dots T(i)$ **do**
9:         $x_a, y_a \leftarrow \text{GetMiniBatch}(\mathcal{D}_A)$;
10:       $x_b, y_b \leftarrow \text{GetMiniBatch}(\mathcal{D}_B)$;
11:       $g \leftarrow \text{TeacherNetBackward}(x_a \cup x_b, y_a \cup y_b; w)$;
12:       $\hat{g} \leftarrow \text{StudentNetBackward}(x_b, y_b; \hat{w})$;
13:       // Teacher update with SGD
14:       $w \leftarrow w - \alpha g$;
15:       // $f$ takes weights, gradients and activations
16:       $\Delta \hat{w} \leftarrow \hat{g} \cdot f(\hat{w}, \hat{g}, \hat{h}; \theta)$;
17:       // Student update with meta-learner $f$
18:       $\hat{w} \leftarrow \hat{w} - \alpha \Delta \hat{w}$;
19:       $h \leftarrow \text{TeacherNet}(x_a \cup x_b; w)$;
20:       $\hat{h} \leftarrow \text{StudentNet}(x_a \cup x_b; \hat{w})$;
21:       $L \leftarrow \text{HuberLoss}(h, \hat{h})$;
22:       $s \leftarrow s + 1$;
23:       // Meta-learner update, backprop thru time $s$ steps
24:       $\theta \leftarrow \theta - \eta \text{T-BPTT}(L, \theta, s)$;
25:       **if** $L > \text{LossThreshold}(i)$ **then**
26:           **break**; // Restart learning
27:       **end if**
28:       **if** $s \geq \text{T-BPTTSteps}(i)$ **then**
29:           $s \leftarrow 0$; // Reset T-BPTT steps
30:       **end if**
31:     **end for**
32: **end for**

The overall algorithm has two nested loops, like many other meta-learning algorithms. The inner loop simulates the experience of doing a single learning process, and the outer loop rolls the network

back to its initial point and re-starts the learning again. Different from standard hyper-parameter optimization, in our proposed algorithm, the meta-learner is updated every step in the inner loop, which makes the training more efficient. The meta-training procedure is detailed in Algorithm 1.

In the beginning of each inner loop, the teacher network and the student network share the same initialization that is pre-trained on Task A. For every step in the inner loop, as illustrated in Figure 2, the teacher network performs a regular SGD update using data from both the old and new tasks. The student network, who does not have access to the old task, computes the gradients on the new task and sends them to the meta-learner network, which then predicts a multiplicative gating. Now using the newly updated weights, we compare the representation difference between the teacher and student networks, and update the meta-learner to minimize this difference.

We use Huber loss multiplied with a scalar hyperparameter as the objective for minimizing the representational differences. To help the meta-learner gradually make progress, we set up a curriculum such that whenever the loss is greater than certain threshold, we will reinitialize the learning process to prevent the meta-learner deviating too far (see Line 25 of Algorithm 1). To speed up learning, we perform truncated backpropagation through time on the meta-learner LSTM. Gradient accumulation is reset whenever the number of unrolled steps is longer than the truncation steps (see Line 28 of Algorithm 1).

## 5 EXPERIMENTS

In this section, we first give implementation details of our algorithm, and then report experimental results on three sets of experiments. In the first set, we test whether our meta-learning algorithm can learn useful learning rules through many sequential training episodes. In the second and third sets of experiments, we verify the generalization ability of our meta-learner, by using unseen classes and unseen initialization checkpoints for evaluation.

### 5.1 IMPLEMENTATION DETAILS

A separate three-layer stacked LSTM meta-learner is learned for each layer of the student network (except the classification layer). The meta-learner uses ReLU activation functions in the hidden layers and tanh in the output layer. The weights and biases of the output layer are initialized to 0 and 1 respectively, to produce a reasonable value at the starting time. A learnable scaling coefficient is then applied to the output to adjust the range. For the classification layer, we apply standard SGD without a meta-learner.

**Meta-learner specification:** For convolutional layers, we take the average over the spatial window of the pre- and post- activations. The convolutional kernel of size $k_H \times k_W \times C_{in}$ is flattened to a vector as the input to the meta-learner, which outputs the weight updates of size $k_H \times k_W \times C_{in}$. For example, for a $3 \times 3 \times 10$ convolutional layer, the input dimension of the meta-learner is $3 \times 3 \times 10$ (weights) $+3 \times 3 \times 10$ (gradients) $+10$ (pre-act) $+ 1$ (post-act) $= 191$, and the output dimension is $3 \times 3 \times 10 = 90$. For other layers, $k_H \times k_W$ is not applicable, and the input is $C_{in} \times 3$ (weight, gradient, pre-act) $+1$ (post-act) and the output is $C_{in}$.

**Baselines:** We compare our proposed method to several baselines:

- **SGD** performs standard SGD on new tasks with the same learning rate as Task A.
- **SGD $\times$ 0.1** is standard SGD with $0.1\times$ learning rate. This is to see whether forgetting on Task A can be traded off with learning progress on Task B.
- **Learning without Forgetting (LwF)** (Li & Hoiem, 2018) distills new data on the old classification branch as additional regularization. We validate the regularization coefficient and set it to 1.0.
- **Elastic Weight Consolidation (EWC)** (Kirkpatrick et al., 2016) adds a quadratic regularizer on the weights, where the regularization strength is computed as a diagonal approximation of the Fisher information matrix. We validate the regularization coefficient and set it to 1.0.

### 5.2 EXPERIMENT 1: SEQUENTIAL LEARNING ON TWO TASKS

We first conduct experiments on MNIST and CIFAR-10 to verify the effectiveness of our meta-learner. To ensure that the meta-learner does not overfit to the training examples, we split the data of Task B into two parts evenly, denoted as $\mathcal{D}_{B_1}$ and $\mathcal{D}_{B_2}$ respectively. At meta-training time, only $\mathcal{D}_{B_1}$ will be used; at meta-test time, only $\mathcal{D}_{B_2}$ will be used.

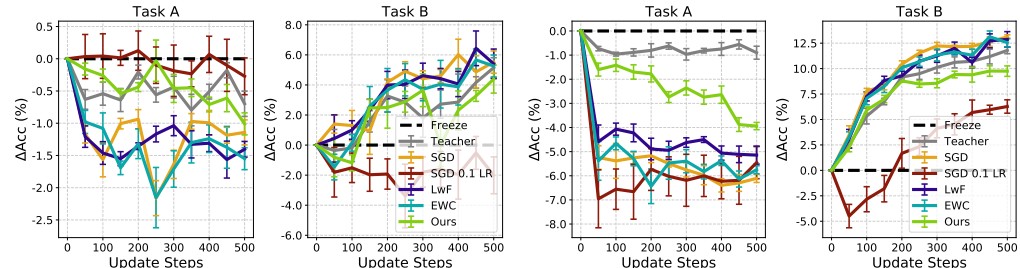

Figure 4: **Left:** Exp 1: MNIST $\mapsto$ FashionMNIST, "Freeze" has 98.37% on Task A and 75.64% on Task B; **Right:** Exp 1: CIFAR 5A $\mapsto$ 5B, "Freeze" has 92.31% on Task A and 74.81% on Task B. Error bar denotes standard error of the mean of 5 runs.

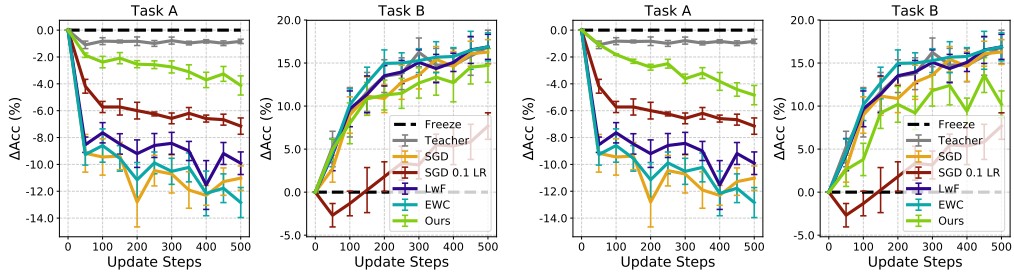

Figure 5: **Left:** Exp 2: CIFAR 5A $\mapsto$ 100 with unseen classes; **Right:** Exp 3: CIFAR 5A $\mapsto$ 100 with unseen initialization checkpoints. "Freeze" has 92.25% on Task A and 71.68% on Task B. Error bar denotes standard error of the mean of 5 runs.

**MNIST $\mapsto$ FashionMNIST:**  The network is first pre-trained on MNIST (Lecun et al., 1998) (i.e., Task A) and then trained on FashionMNIST (Xiao et al., 2017) (i.e., Task B). The main network contains two 5x5 conv layers and two fully-connected layers with dimensions [20, 50, 500, 10], respectively, and the meta-learner is a 3-layer LSTM with 32 hidden unit. SGD with learning rate 1e-2 and momentum of 0.5 is used throughout pre-training. During meta-learning, we use learning rate 5e-2 without momentum. We set the curriculum threshold to 20 and T-BPTT step to 5 initially and increase them by 5 and 2 every 1000 episodes until 35 and 11, respectively. The loss coefficient is set to 300. We train the meta-learner using the Adam optimizer with learning rate 1e-3 for a total of 4000 episodes. At meta-test time, we take the activations before the last layer and train a linear readout layer with 100 Adam optimization steps using learning rate 1e-1.

**CIFAR-10:**  We split the CIFAR-10 dataset into two subsets, the first subset (CIFAR-5A) consists of the first 5 classes ("airplane", "automobile", "bird", "cat", "deer") and the second subset (CIFAR-5B) consists of the remaining 5 classes ("dog", "frog", "horse", "ship", "truck"). A ResNet-32 (He et al., 2016) network is first pre-trained on CIFAR-5A, with all BatchNorm (Ioffe & Szegedy, 2015) layers replaced by GroupNorm (Wu & He, 2018), using a learning rate of 0.1 and momentum of 0.9. During meta-learning, we use learning rate 0.1 without momentum. For pre-training we use 128 examples as a mini-batch, and for meta-learning we use 128 for the teacher and 64 for the student (Task B only). The meta-learner is a 3-layer LSTM with 64 hidden units for the convolutional kernels and 32 hidden units for $\gamma$ or $\beta$ of the GroupNorm layers. We set the curriculum threshold to 20 and T-BPTT step to 5 initially and increase them by 5 and 2 every 300 episodes until 30 and 9, respectively. The loss scaling coefficient is set to 300. We train the meta-learner using the Adam optimizer with learning rate 1e-3 for a total of 900 episodes. At meta-test time, we take the activations before the last layer and train a linear readout network using 500 Adam optimizer steps with a learning rate of 1e-1.

**Results:**  Figure 4 shows results on MNIST $\mapsto$ FashionMNIST and CIFAR-10. The error bar is the standard error of the mean of 5 runs. All entries in the table use the proposed readout measure. We found that reducing learning rate cannot help prevent from catastrophic forgetting. Our meta-learner outperforms other methods by a large margin on Task A. On Task B, our meta-learner has very similar performance to the teacher network, which matches our expectation.

### 5.3 EXPERIMENT 2: GENERALIZING TO UNSEEN CLASSES

In Experiment 1, Task B is the same for both training and testing. To verify the generalization ability of our model to unseen classes, we utilize the CIFAR-100 dataset, to construct two different tasks, $B_1$ and $B_2$ from disjoint subset of data. We start from an initial model trained on CIFAR-5A, at meta-training time we train a meta-learner on Task $B_1$, while at meta-test time we evaluate meta-

|        | Exp 1 MNIST/Fashion | Exp 1 CIFAR-5A/5B | Exp 2 CIFAR-5A/100 | Exp 3 CIFAR-5A/100 |
|--------|---------------------|-------------------|--------------------|--------------------|
| Freeze | 84.8 | 65.1 | $68.6 \pm 2.9$ | $68.6 \pm 2.9$ |
| SGD | $86.8 \pm 1.0$ | $68.7 \pm 1.5$ | $71.0 \pm 2.1$ | $71.0 \pm 2.1$ |
| SGD $\times 0.1$ | $85.0 \pm 0.7$ | $65.4 \pm 2.1$ | $69.7 \pm 2.3$ | $69.7 \pm 2.3$ |
| LwF | $87.2 \pm 1.1$ | $69.3 \pm 0.7$ | $73.1 \pm 1.7$ | $73.1 \pm 1.7$ |
| EWC | $86.9 \pm 1.7$ | $67.7 \pm 1.0$ | $72.9 \pm 2.1$ | $72.9 \pm 2.1$ |
| Ours | $\mathbf{87.3} \pm 0.7$ | $\mathbf{70.5} \pm 1.1$ | $\mathbf{75.0} \pm 1.3$ | $\mathbf{73.6} \pm 2.6$ |
| Teacher | $88.0 \pm 1.1$ | $76.7 \pm 0.3$ | $81.9 \pm 1.2$ | $81.9 \pm 1.2$ |

Table 1: Joint task performance on Experiment 1, 2 & 3, after 500 update steps. $\pm$ denotes standard error of the mean of 5 runs.

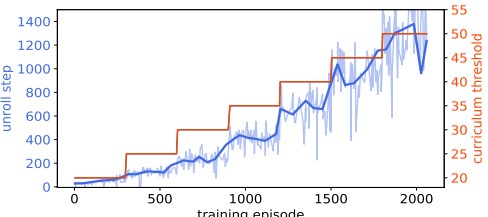

Figure 6: Training curve on Experiment 2

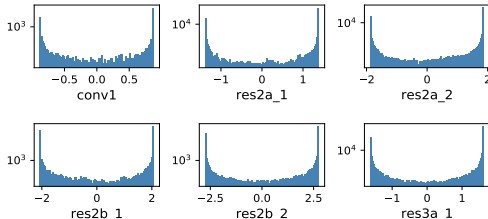

Figure 7: Visualization of the meta-learner outputs

learner on Task $B_2$. Unlike the previous experiment, the class definition changes from meta-training to meta-testing. This is a more practical setting since in reality we do not know a priori which new task the model needs to adapt to.

**CIFAR-100:** We split CIFAR-100 dataset into two subsets. The first subset consists of the first 50 classes, the second subset consists of the remaining 50 classes. At meta-training time we randomly sample 5 classes from the first subset to constitute Task $B_1$ at the beginning of every episode. And at meta-test time, we randomly sample 5 classes from the second subset to constitute Task $B_2$, and we repeat it for 5 times to take the average performance. Task $B_1$ and Task $B_2$ have no overlap for both images and classes. We set the curriculum threshold to 20 and the T-BPTT step to 5 initially and increase them by 5 and 2 every 300 episodes until 50 and 17 and train the meta-learner for a total of 2k episodes. Figure 6 illustrates the curriculum threshold schedule and number of unrolled steps to indicate the training progress. Other hyper-parameters are keep the same as CIFAR-10 in this experiment.

**Results:** As shown in Figure 5, our meta-learner generalizes well to unseen classes and clearly outperforms other baselines. By the end of 500 update steps, the representation forgetting on Task A is 4% for our model, compared to over 10% for SGD, EWC, and LwF; meanwhile on Task B our model performs much than SGD $\times 0.1$, only $\approx 2\%$ behind other baselines.

### 5.4 EXPERIMENT 3: GENERALIZING TO UNSEEN INITIALIZATION

In Experiment 3, we remove the assumption on a fixed initialization checkpoint. In order for the meta-learner to generalize to different initialization state, during training time we provide 100 different pretrained checkpoints and each training episode uses a different checkpoint. At test time, an unseen checkpoint is provided.

**Results:** As shown in Figure 5, our meta-learner generalizes well to unseen initialization (as well as unseen classes) and still outperforms other baselines. The performance drops a little as expected, since the meta-learner model has to learn to adapt different initializations during meta training/test.

**Joint task performance:** We include the joint task performance in Table 1 for all Experiments 1-3, where we train a joint classifier on the representations learned after 500 update steps on Task B. It can be confirmed that the sequentially learned representations produced by the meta-learner are useful to both tasks, outperforming other sequential learning baselines.

### 5.5 VISUALIZATION OF META-LEARNER OUTPUTS

To further understanding the behavior of our meta-learner, we visualize the distribution of the output of the meta-learner (i.e. the gradient multiplier $\delta$) in Figure 7. Our meta-learner produces non-trivial outputs that are not simply a global scaling of the learning rate. Sometimes the multiplier can be negative, which means the final update direction is opposite to the gradient descent direction. It shows that the meta-learner can learn to dynamically modify the gradient direction to prevent catastrophic forgetting.

## 6 CONCLUSION AND FUTURE WORK

Catastrophic forgetting handicaps state-of-the-art deep neural networks from learning online tasks in the wild. This paper studies the effect of representational forgetting in a sequential learning framework. In particular, we propose to add a linear readout layer to test the amount of forgetting at the representation level, where a significant drop in performance on old tasks is still observed, consistent with prior literature. We then propose to train a meta-learner to predict the weight updates, with supervision from a multi-task teacher network. Our meta-learner is able to overcome catastrophic forgetting while improving its performance on new tasks. We further verify that our meta-learner has the ability to generalize to unseen classes and checkpoint initializations. Currently we have made the meta-learner successful at predicting weight updates for up to 500 steps, but we still find it challenging to let it generalize to even longer sequences. In the future, we expect these issues can be addressed by potentially training the meta-learner with longer sequences and more sequential tasks with more computational time, and combining meta-learning techniques for dealing with longer horizon (e.g. Liao et al. (2018); Metz et al. (2019)).

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
