# OpenReview forum: "Learning to Remember from a Multi-Task Teacher"
_ICLR.cc/2020/Conference — Reject_

### Official Review · AnonReviewer3 · 2019-10-22
**Official Blind Review #3**

**Rating:** 3

**Review:**

###  Summary

- The paper demonstrates that neural networks that appear to have forgotten an old task still contain useful information of that task in their representation layers.
- The paper proposes to meta-learn an update rule (parameterized by an LSTM) that acts as a gating mechanism (or plasticity) for each learnable parameter at meta-test time.
- To meta-learn the update rule, the paper proposes minimizing the difference between representations of a teacher and student neural network. The teacher neural network learns from a batch of data sampled IID from the distribution of the complete dataset whereas the student neural network samples a batch only from the current task.

### Decision with reasons

I vote for rejecting the paper.

1- The claim that neural networks forget mostly due to a miscalibration of the output layer is not well supported empirically (The drop in readout accuracy in Figure 1 is still significant). If the claim is only to the extent that the drop in readout accuracy is slower than original accuracy, then it's not interesting or new. (This is what I believed in before reading the paper as well).

2- While the underlying idea in the paper for learning an update rule is promising and sound, the paper is missing baselines that also use the meta-training dataset in some way. Moreover, a meta-learned update rule is only useful if it can discover some general underlying learning principles. In this paper, the meta-train and meta-test settings are too similar to see if that is the case.


### Supporting arguments for the reasons for the decision.

1- The paper claims that catastrophic forgetting in a neural network is partly due to miscalibration of the last layer, and the representation layer of the neural network still contain useful information. However, the only supporting evidence for this claim is that readout accuracy does not drop as quickly as the original accuracy (Figure 1).

First, the drop in readout accuracy is still significant to term forgetting 'catastrophic.' Secondly, figure 1 only report results after 300 steps. A more interesting question is the difference between the accuracies when the network has been trained on Task B till convergence. Secondly, it is important to report the read-out accuracy for task A on a random Neural Network of the same architecture to see if the Neural network is maintaining information in the representation layer (as the authors claim), or if a linear classifier on a random CNN is just a strong baseline (Shown to be a strong baseline in many recent papers. One example is Anand et.al 2019 [1])

2- The motivation behind meta-learning an update rule is to discover underlying learning principles that generalize to new settings. Metz et. al. 2019, for example, showed that their learned update rule could be applied to networks with different architecture, non-linearities, and datasets (They went as far as showing it worked on different data modalities.)

All the results in this paper, however, are for a fixed architecture (The authors do look at generalization to unseen classes, but we care about generalization to arbitrary architectures/problems when meta-learning an update rule). The data at meta-train and meta-test time are also very similar (Different parts of the same dataset). The empirical results, consequently, are not very convincing. Moreover, by reading between the lines, it can be inferred that the learned update rule is very finicky. For instance, to generalize just to unseen initializations, the authors had to use 100 different initializations at meta-training time. That does not instill a lot of confidence in me about the stability of the learned update rule.

Finally, the paper proposes the complex student-teacher learning paradigm while skipping a simple baseline: training on the student model by using data from Task B in the support set and using data from A and B in the query set during meta-training. A similar procedure was proposed by Javed and White 2019 [3]. (Note that the current baselines in the paper do not use the meta-training data at all which makes the comparison extremely unfair. Moreover, even a simple baseline such as LwF that does not use meta-training performs almost as well (See Table 1).)

### Additional evidence that can change my evaluation

1- Showing that the meta-learned update rule can be applied to different architectures/non-linearities/datasets (Train on one dataset, test on another).

### Minor comments that did not play a part in my decision, but should be addressed nonetheless.

The paper should cite the classic paper by Yoshua. et.al (1991) which proposed the idea of meta-learning an update rule [2].

The paper, in its current form, needs to be proofread and reorganized. There are many errors in the grammar (For example just in the first paragraph, New borns -> Newborns, a same -> the same (or 'a distribution')). I find passing my writing through the free version of Grammarly very helpful in getting rid of most such errors.

The organization of the paper is also not very clear. For example, the third paragraph in "Related Work" is about the method proposed in the paper whereas the second and fourth are about related work.

The writing is also occasionally ambigious. For instance:

"In human language acquisition, it is found that children who lost their first language maintain similar brain activation to bilingual speakers (Pierce et al., 2014).

Inspired by this fact, we propose a novel meta-learning algorithm that tries to mimic a multi-task teacher network’s representation, an offline oracle in our sequential learning setup, since multi-task learning has simultaneous access to all tasks whereas our sequential learning algorithm only has access to one task at a time."

It is not clear how the method in the second paragraph is inspired from the statement in the first paragraph.

I did not take writing quality in account when giving my score because openreview allows updating the paper during the review process. I hope that authors would fix these issues during the writing process.

On an unrelated note, the figures in the paper are well made and clear.  It is possible to understand the proposed methodology just from the figures.

[1] Unsupervised State Representation Learning in Atari https://arxiv.org/abs/1906.08226

[2] Learning a Synaptic Learning Rule https://mila.quebec/wp-content/uploads/2019/08/bengio_1991_ijcnn.pdf

[3] Meta-Learning Representations for Continual Learning https://arxiv.org/abs/1905.12588


#### UPDATE
I gave the paper a 3 for the following reasons:

1. The baselines do not use the meta-training dataset at all. This makes the comparisons unfair (Is the update rule learning some general learning principles or learning or induce good representations for the meta-training dataset?)

2. The meta-train and meta-test settings are too similar. Learning an update rule only makes sense if we can discover some underlying learning principles. If the update rule is tied to a data-distribution, it is not extremely useful.

In the public discussion phase, the authors addressed both of my concerns. They added baselines which uses the meta-training dataset for learning a representation, and they added an experiment in which meta-testing is done on a different dataset. However:

1. The baselines perform very close to the proposed method at a fraction of meta-training cost.  Rep*, even before doing any steps on Task B, results in better average performance than the method proposed in the paper. Moreover, the authors do not combine these baselines with existing methods to mitigate interference (Such as LwF) which can easily be done (and would probably increase the performance of the baseline noticeably)

2. The meta-training and meta-testing datasets are still too similar in the new experiment (A downscaled TinyImagenet is very similar to CIFAR). Even though the results are more promising given the added experiment, I don't think they answer if the LSTM is learning some general learning principle or some task-specific heuristic for performing slightly better.

R2's review (and the response to the review) also highlights an important problem -- the authors didn't tune the optimizers used by the baselines on the meta-training dataset. I suspect that a well-tuned adaptative optimizer (Like Adam) would reduce the gap between baselines and the author's method significantly.

Is hundreds-of-hours of GPU compute worth negligible (or no) performance improvement in a very restricted setting (Since the learning rule doesn't seem to generalize based on existing results)? I'm inclined to say that it is not. As a result, I'm keeping my initial score.

I do encourage the authors to investigate the proposed method more (It is a reasonable method) and try to empirically demonstrate that the LSTM is discovering some general learning principle.

**Experience Assessment:**

I have published one or two papers in this area.

**Review Assessment: Checking Correctness Of Derivations And Theory:**

N/A

**Review Assessment: Checking Correctness Of Experiments:**

I carefully checked the experiments.

**Review Assessment: Thoroughness In Paper Reading:**

I read the paper thoroughly.

---

> ### Author Response · Authors · 2019-11-14
> **Response to R3**
>
> We thank R3 for the insightful and constructive feedback. We address individual comments below.
>
> ------
> Re: Significance of Figure 1: The original Figure 1 has a different setting compared to the main experiment, where it was only a binary classification problem, so chance is 50% and a drop of 25% is very significant. We agree with R3 that this point is not clearly illustrated, so since then we have updated the Figure 1 to use the same 5-way classification setting in the main experiment. As illustrated, regular SGD not only has a severe impact on the output layer, but also has a very large variance across runs, whereas in readout training, the level of forgetting is much more controlled and less noisy. Therefore, we use Figure 1 to argue that using readout training to measure the level of representational forgetting is a better measure.
>
> ------
> Re: Transfer to less similar datasets: Thanks for pointing out. To illustrate more transferability settings, we use 5 random classes of TinyImageNet as Task B, and we report the results in Table-1. Note that the meta-network is pretrained on CIFAR-100, and has not seen a single image in Tiny-ImageNet. As shown in Table-1, Our meta-learner can preserve the most information in Task A, while achieving similar performance as Teacher on Task B.
> Table-1: Results on CIFAR 5A -> Tiny ImageNet
>           |     Task A     |     Task B     |
> ---------------------------------------------
> 0 step of training Task B
> ---------------------------------------------
> Freeze    |  92.4 +/- 0.2  |  65.5 +/-  4.4 |
> ---------------------------------------------
> 500 steps of training Task B
> ---------------------------------------------
> SGD       |  78.3 +/- 2.5  |  77.0 +/- 1.6  |
> SGD x0.1  |  87.9 +/- 0.4  |  72.1 +/- 2.6  |
> LWF       |  81.3 +/- 1.6  |  78.6 +/- 2.3  |
> EWC       |  79.6 +/- 1.6  |  79.2 +/- 1.7  |
> Ours      |  87.2 +/- 0.6  |  75.5 +/- 1.9  |
> ---------------------------------------------
> Teacher   |  89.3 +/- 0.5  | 76.0 +/- 3.1   |
>
> -------
> Re: Baselines: We thank R3 for pointing out these baselines. During the past few days, we gathered initial experimental results of three new baselines, as requested by the reviewer.
>
> - Random: This is a baseline suggested by R3. We use random feature projection of the images using ResNet-32 and learn a linear readout layer.
>
> - Rep: This is representation learning on the meta-training set, as pointed out by R3 we don’t have a baseline that leverage meta-training data. We use the meta-training 50 classes plus CIFAR-5A original classes to pre-train a representation backbone using standard classification. Then we use linear readout to directly get the classification accuracy of old+new classes.
>
> - MAML (Finn et al., 2017): We first pretrain with CIFAR-5A, then for each meta-learning step, we unroll the readout SGD steps for 10 steps, and then backprop through SGD from the “query” set to learn a good representation of the backbone. During testing, we train the readout layer till convergence. Note that this is an attempt to replicate the MAML method proposed in Javed and White (2019) in our experimental settings. The OML method cannot be adapted to our setting, since we do many SGD steps per task instead of one per task.
>
> As shown in Table-2, even though learning a good representation on 50+5 classes can contribute some gain over the “Freeze” baseline, finetuning representation on the new classes are still necessary. Furthermore, Rep and MAML still suffer from catastrophic forgetting on the representation level on Task A.
>
> Table-2: Additional baselines for Experiment 2
>           |     Task A     |     Task B     |
> ---------------------------------------------
> 0 step of training Task B
> ---------------------------------------------
> Freeze    |  92.4 +/- 0.2  |  71.1 +/- 7.5  |
> Random    |  24.6 +/- 3.5  |  34.3 +/- 7.2  |
> Rep*      |  93.3 +/- 0.6  |  82.6 +/- 4.6  |
> MaML*     |  93.6 +/- 0.2  |  80.7 +/- 4.7  |
> ---------------------------------------------
> 500 steps of training Task B
> ---------------------------------------------
> SGD       |    78.4 +/- 1.9   |  88.2 +/- 1.3  |
> SGD 0.1 |    84.8 +/- 0.5   |  81.5 +/- 2.5  |
> LWF       |    81.8 +/- 2.0   |  89.5 +/- 1.1  |
> EWC       |    80.6 +/- 0.8  |  88.8 +/- 1.1  |
> Rep*      |  80.2 +/- 2.1 |  87.9 +/- 3.1  |
> MaML*     |  82.4 +/- 2.6  |  89.1 +/- 3.7  |
> Ours^      |  88.0 +/- 0.6  |  86.7 +/- 1.3  |
> ---------------------------------------------
> Teacher    |  91.1 +/- 0.4  |  88.9 +/- 1.4  |
> *: Main network parameters are pre-trained with all 50+5 meta-training classes instead of 5 classes.
> ^: Meta-learner parameters are trained with 50 meta-training classes, but the main network parameters are only trained with 5 classes.

---

> > ### Author Response · Authors · 2019-11-14
> > **Response to R3 (part 2)**
> >
> > ------
> > Re: Reference: We thank R3 for pointing out the prior literature (Yoshua et al., 1991) and we have already updated our manuscript and cited this influential work.
> >
> > ------
> > Re: Writing: We thank R3 for pointing out. We have already updated our manuscript and fixed these typos. We will do another pass to make sure the writing is clear and reorganizing the paragraphs.
> >
> > ------
> > References:
> > Chelsea Finn, Pieter Abbeel, and Sergey Levine. "Model-agnostic meta-learning for fast adaptation of deep networks." In ICML 2017.
> > Khurram Javed, Martha White. “Meta-Learning Representations for Continual Learning.” In NeurIPS 2019 (to appear).

---

### Official Review · AnonReviewer2 · 2019-10-23
**Official Blind Review #2**

**Rating:** 8

**Review:**

Summary:
This paper explores learning without forgetting / the online learning setting. They employ a novel meta-learned learning algorithm to this end.

Writing:
For the most part the writing was clear and easy to follow. There where a couple typos on the top of page 2 that should be fixed.

Motivation:
The motivation for wanting meta-learning as well as various algorithmic choices are clear.
The one piece of motivation I did not fully understand is why not forgetting on the feature space is so important. My understanding of the method is that it should be applicable in both settings (with and without relearning the last layer). Infact, I would expect the difference between the meta-learned method and the baselines to only increase in this setting.

I find the distillation based learning to be a clever alternative to the computationally heavy optimizing over past performance.

Experiments:
This work provides a nice build up of experiments.
Experiment 1 demonstrates the principles. In my opinion you should caution the reader given the meta-train, meta-test split. D_{B_1} and D_{B_2} are the same distribution and thus it will be easy for the learned update rule to memorize features. Given your learned update rule
architecture I doubt this will be the case though. I believe the authors are aware of this though as this issue is addressed in experiments 2 and 3.

Please include what the error bars are over in the captions.

Experiments 2 and 3 are interesting and demonstrate the method on a more realistic setting. From the details it seems like this was difficult to get to work -- needing a complex schedule for example. Further elaboration or study of these details (e.g. ablations) would help the field. Also please include what the +- is for the experiments in table 1.

Figure 6 is not referenced in the text. It was also difficult for me to understand though I finally got it.

Overall, I believe the baselines could be made considerably stronger. Meta-learning expends considerable compute to find a good learned update rule. Spending similar amounts of compute tuning the baselines would be appreciated. Second, the meta-learned update rule presented here is essentially a learned optimizer and thus considerably more powerful than SGD. What optimizers did you use for LwF and EWC? Where the hyper parameters tuned here in an attempt to use similar compute? Where there learning rate schedules also tuned?

Questions / concerns:

Cost of running this not discussed. I would expect that both meta-training, and training are considerably more expensive. I am curious in particular

One motivation for meta-learning update rules in this way is that this cost can be amortized ahead of time and the learned update rule can transfer to new very different tasks. Without transfer like this, however, it's unclear if a method such as this is useful in general. Some discussion to this end I think would be helpful. I am not docking this work for not doing this type of generalization work though as we must start someplace and meta-training on similar data distributions is a logical place to do so.

I am unclear as to your exact meta-training setup from algorithm 1. Does your meta-gradient (DL/dtheta) get computed every inner iteration (iteration of t)? If so how many steps do you back prop through? As of now it looks like your only backpropping a single iteration / application of f. Second, when computing this meta-gradient do you compute the true derivative or a first order approximation common in other work?

Overall:
I would recommend this paper for acceptance as it presents an interesting approach to solving the catastrophic forgetting issue with a compelling set of diverse experiments.


**Experience Assessment:**

I have read many papers in this area.

**Review Assessment: Checking Correctness Of Derivations And Theory:**

N/A

**Review Assessment: Checking Correctness Of Experiments:**

I carefully checked the experiments.

**Review Assessment: Thoroughness In Paper Reading:**

N/A

---

> ### Author Response · Authors · 2019-11-14
> **Response to R2**
>
> Thank you for your valuable review. We provide detailed response below.
>
> -------
> Re: Figure 6: We have referenced Figure 6 in the main text and updated the error bar information in table and figure captions.
>
> -------
> Re: Optimizers and hyperparameters tuning: All entries including baselines as well as ours use SGD optimizer and learning rate 0.1 which follows the standard CIFAR-10 training setting that has a decent classification performance. For the loss scaling and curriculum threshold, we have explored a few settings to make the number of unrolling step to be gradually increasing, but we did not extensively tune the schedule.
>
> -------
> Re: Cost of running: It took ~100 GPU hrs to train a meta learner for 500 unrolled steps and we let the model kept running for a total of 600 GPU hours for 1200 unrolled steps. Note that our current implementation may not be optimized and can be made faster.
>
> -------
> Re: transfer to new tasks: Thanks for pointing out. To illustrate more transferability settings, we use 5 random classes of TinyImageNet as Task B, and we report the results in Table-1. Note that the meta-network is pretrained on CIFAR-100, and has not seen a single image in Tiny-ImageNet. As shown in Table-1, Our meta-learner can preserve the most information in Task A, while achieving similar performance as Teacher on Task B.
>
> Table-1: Results on CIFAR 5A -> Tiny ImageNet
>           |     Task A     |     Task B     |
> ---------------------------------------------
> 0 step of training Task B
> ---------------------------------------------
> Freeze    |  92.4 +/- 0.2  |  65.5 +/-  4.4 |
> ---------------------------------------------
> 500 steps of training Task B
> ---------------------------------------------
> SGD       |  78.3 +/- 2.5  |  77.0 +/- 1.6  |
> SGD x0.1  |  87.9 +/- 0.4  |  72.1 +/- 2.6  |
> LWF       |  81.3 +/- 1.6  |  78.6 +/- 2.3  |
> EWC       |  79.6 +/- 1.6  |  79.2 +/- 1.7  |
> Ours      |  87.2 +/- 0.6  |  75.5 +/- 1.9  |
> ---------------------------------------------
> Teacher   |  89.3 +/- 0.5  | 76.0 +/- 3.1   |
>
> -------
> Re: meta-gradient computed every inner iteration: Yes we compute meta-gradient for every inner iteration. We maintain a maximum step T for the steps that we backprop through, and T gradually increase throughout training following a curriculum. This is mentioned at the end of the model section as well as in each experiment subsections.
>
> -------
> Re: true derivative or first order approximation: We compute the true derivative since this is the definition for gradients. We acknowledge that there are prior literature that shows first order approximation can also work (e.g. “MAML” and “learning to learn by gradient descent by gradient descent”).

---

### Official Review · AnonReviewer1 · 2019-10-24
**Official Blind Review #1**

**Rating:** 1

**Review:**

Summary: This paper introduces a variation on measuring catastrophic forgetting in sequential learning at the representation level and attempts to resolve forgetting issue with the help of a meta-learner that predicts weight updates for previous tasks while it receives supervision from a multi-task learner teacher. The new method is evaluated on sequences of two tasks while task 1 data remains available at all times to the teacher.

Pros:
(+): This paper is very well-written and very well-motivated.
(+): Tackling continual learning from a meta-learning approach is novel and not yet well-explored.
(+): Literature review is done precisely well.

Cons that significantly affected my score and resulted in rejecting the paper are two-fold.

First, based on my understanding from the paper, it appears that this work has a significant contradictory assumption with a regular continual learning setup and that is to provide access to the entire dataset from an old task while we learn a new task. This changes the problem from continual/sequential/lifelong learning to multi-task learning. All the prior work that were beautifully reviewed in section 1 and 2 obey this assumption where access to previous tasks’ data is either impossible (ex. [1,3,4,5,6,7,8] in the below list ) or is very limited (ex. [2]).

Second, is the experimental setting. The experiments are accurately described and performed but authors have only considered sequence of 2 tasks which is far from being considered as a continual learning setting. I would like to ask the authors to explain how this method can be extended to multiple tasks and how much of the past data they should provide while training? Another drawback in the experiments is about the baselines. Despite addressing the most recent papers in section 2, authors have only made comparison against two relatively old approaches (EWC by Kirkpatrickthat et al from 2016 as well as LwF by Li & Hoiem presented at ECCV 2016, I believe the authors have cited the journal version of the work published in 2018 but the work is actually from ECCV 2016). Although these methods are still included as baselines in the literature, more recent approaches which have outperformed these need to be provided as well. I have provided a list of papers which achieved superior performance to the current baselines below which is arranged chronologically and is indeed not limited to this list as it is not realistic to list all prior work since 2016 in here.

I would be happy to change my score if authors can address the above concerns about considering distinguishing multi-task learning from continual learning and providing a realistic evaluation setup with more than 2 tasks and comparison with current state of the art methods.

[1] Zenke, Friedemann, Ben Poole, and Surya Ganguli. "Continual learning through synaptic intelligence." Proceedings of the 34th International Conference on Machine Learning-Volume 70. JMLR. org, 2017.
[2] Lopez-Paz, David, and Marc'Aurelio Ranzato. "Gradient episodic memory for continual learning." Advances in Neural Information Processing Systems. 2017.
[3] Shin, Hanul, et al. "Continual learning with deep generative replay." Advances in Neural Information Processing Systems. 2017.
[4] Nguyen, Cuong V., et al. "Variational continual learning." arXiv preprint arXiv:1710.10628 (2017).
[5] Serrà, J., Surís, D., Miron, M. & Karatzoglou, A.. (2018). Overcoming Catastrophic Forgetting with Hard Attention to the Task. Proceedings of the 35th International Conference on Machine Learning, in PMLR 80:4548-4557
[6] Schwarz, Jonathan, et al. "Progress & compress: A scalable framework for continual learning." arXiv preprint arXiv:1805.06370 (2018).
[7] Mallya, Arun, and Svetlana Lazebnik. "Packnet: Adding multiple tasks to a single network by iterative pruning." Proceedings of the IEEE Conference on Computer Vision and Pattern Recognition. 2018.
[8] Ebrahimi, Sayna, et al. "Uncertainty-guided Continual Learning with Bayesian Neural Networks." arXiv preprint arXiv:1906.02425 (2019).
[9] Aljundi, Rahaf, et al. "Online continual learning with no task boundaries." arXiv preprint arXiv:1903.08671 (2019).

------------------------------------------------------------------------------------------------------------------------------------------------------
------------------------------------------------------------------------------------------------------------------------------------------------------
------------------------------------------------------------------------------------------------------------------------------------------------------
POST-REBUTTAL review:
I disagree with the authors claiming that this work is continual learning (sequential learning + avoiding forgetting).
Despite introducing 9 recent continual learning work to authors in my initial review, they added 2 meta-learning baselines (MAML,REP), keeping 2 naive and old CL baselines is not acceptable. I reply to authors comment below regarding the baselines:

[Authors' reply:] Lopez-Paz, David, and Marc'Aurelio Ranzato. "Gradient episodic memory for continual learning." and Shin, Hanul, et al. "Continual learning with deep generative replay." and Nguyen, Cuong V., et al. "Variational continual learning” and Aljundi, Rahaf, et al. "Online continual learning with no task boundaries." We argue that these paper have a different setting compared to ours since they require a buffer whereas our method has no storage of past data/gradient. Having past data storage can usually improve the performance and our method can potentially also get a boost. Having a data buffer can also cost a lot of memory storage depending on input/weight dimension. Therefore, we argue it won’t be a fair setting to compare with methods with data buffers.

[Reviewer's reply:] GEM (Lopez-Paz et al., 2017) and its faster version (A-GEM) (Chaudhry, et al. 2018) and other memory based methods  such as MER (Riemer et al. 2018), ER-RES (Chaudhry et al. 2019), they use memory sizes of at most 6MB to store samples but they only do a **single epoch** through the data. So if it is not fair, it would be for those methods given the computational expenses of this paper. VCL (Nguyen et al. 2018) in its vanilla version does not use coreset memory if that is still your concern.

-------------------------------------------------------------------------------------------------------------------------------------------------------------------------------------------------------
[Authors' reply:] Zenke, Friedemann, Ben Poole, and Surya Ganguli. "Continual learning through synaptic intelligence." has very similar performance to EWC in their paper. We have cited this work already.

[Reviewer's reply:] This method is an online version of EWC which is faster despite the on-par performance. So it has its own advantage.

--------------------------------------------------------------------------------------------------------------------------------------------------------------------------------------------------------
[Authors' reply:] Serrà, J., Surís, D., Miron, M. & Karatzoglou, A.. (2018). Overcoming Catastrophic Forgetting with Hard Attention to the Task and other pruning based papers. Thanks for pointing out. We have cited and will compare to them in the future. One thing to note is that, in these works, the model needs to know which task ID it is currently dealing with, and thus can turn on the pruning procedure for the next session. This can potentially be a limitation for dynamic incoming tasks.

[Reviewer's reply:] Comparing with HAT paper (Serrà et al. 2018) is really easy using their provided code and is one of the strongest baselines in continual leaning literature. They do NOT do any pruning. Their approach simply learns an attention mask which regularizes weights and prevent changes on them without using any memory. Regarding the task number, this is indeed not an issue for your approach with 2 tasks.

--------------------------------------------------------------------------------------------------------------------------------------------------------------------------------------------------------

[Authors' reply:] Ebrahimi, Sayna, et al. "Uncertainty-guided Continual Learning with Bayesian Neural Networks." has not been published in a conference venue, and it may be too early to compare with it.

[Reviewer's reply:] I agree that this work is not published and hence can't be asked for comparison but I encourage authors to read it.

--------------------------------------------------------------------------------------------------------------------------------------------------------------------------------------------------------
Authors are neglecting one important difference between meta-learning and continual learning: in MAML and REP, it is assumed that we have access to ALL tasks distributions from which we sample from in the beginning (look at page 3, algorithm 1, line 3). This is in contrast to continual learning where one cannot even assume how many tasks will be given. Moreover, the computational expense of this work which causes performing more than 2 tasks to be a future work is also not acceptable when there are significantly cheaper and are able to do a lot more than 2. (In all the references I mentioned, the length of the sequence in experiments is at least 5.)

While this work might be interesting to meta-learning community, I think it is far from being introduced as a method that prevents catastrophic forgetting and hence be included in the CL literature. Therefore, I intend to keep my score as reject.


**Experience Assessment:**

I have published one or two papers in this area.

**Review Assessment: Checking Correctness Of Derivations And Theory:**

N/A

**Review Assessment: Checking Correctness Of Experiments:**

I carefully checked the experiments.

**Review Assessment: Thoroughness In Paper Reading:**

I read the paper thoroughly.

---

> ### Author Response · Authors · 2019-11-14
> **Response to R1**
>
> We thank R1 for valuable and constructive feedback. We address individual points below.
>
> -------
> Re: Usage of old data: R1 points out that our setting has access to the entire dataset from an old task while learning new tasks, and thus has an unfair advantage compared to baselines. We apologize if our writing is unclear, but we believe there is a misunderstanding. The only time when the use of old data happens is during the linear readout stage, which is applied on all baselines. During meta-training we use the old data only for training the teacher network, but the teacher network is NOT used in meta-testing, and only the meta-learner is updating the network alone. Therefore our model is strictly doing sequential learning during meta-test time.
>
> -------
> Re: Sequence of more tasks: Learning more tasks in a sequence would require longer unrolling of steps in the inner loop training to the order of a few thousands. This will be our future work to extend the algorithm towards multiple sequential tasks.
>
> -------
> Re: Baselines and references: To address both R1 and R3’s comment, we added three more baselines.
>
> - Random: This is a baseline suggested by R3. We use random feature projection of the images using ResNet-32 and learn a linear readout layer.
>
> - Rep: This is representation learning on the meta-training set, as pointed out by R3 we don’t have a baseline that leverage meta-training data. We use the meta-training 50 classes plus CIFAR-5A original classes to pre-train a representation backbone using standard classification. Then we use linear readout to directly get the classification accuracy of old and new classes.
>
> - MAML (Finn et al., 2017): We first pretrain with CIFAR-5A, then for each meta-learning step, we unroll the readout SGD steps for 10 steps, and then backprop through SGD from the “query” set to learn a good representation of the backbone. During testing, we train the readout layer till convergence. Note that this is an attempt to replicate the MAML method proposed in Javed and White (2019) in our experimental settings. The OML method cannot be adapted to our setting, since we do many SGD steps per task instead of one per task.
>
> As shown in Table-1, even though learning a good representation on 50+5 classes can contribute some gain over the “Freeze” baseline, finetuning the representation on the new classes is still necessary. Furthermore, Rep and MAML still suffer from catastrophic forgetting on the representation level on Task A.
>
> Table-1: Additional baselines for Experiment 2
>           |     Task A     |     Task B     |
> ---------------------------------------------
> 0 step of training Task B
> ---------------------------------------------
> Freeze    |  92.4 +/- 0.2  |  71.1 +/- 7.5  |
> Random    |  24.6 +/- 3.5  |  34.3 +/- 7.2  |
> Rep*      |  93.3 +/- 0.6  |  82.6 +/- 4.6  |
> MaML*     |  93.6 +/- 0.2  |  80.7 +/- 4.7  |
> ---------------------------------------------
> 500 steps of training Task B
> ---------------------------------------------
> SGD       |    78.4 +/- 1.9   |  88.2 +/- 1.3  |
> SGD 0.1 |    84.8 +/- 0.5   |  81.5 +/- 2.5  |
> LWF       |    81.8 +/- 2.0   |  89.5 +/- 1.1  |
> EWC       |    80.6 +/- 0.8  |  88.8 +/- 1.1  |
> Rep*      |  80.2 +/- 2.1 |  87.9 +/- 3.1  |
> MaML*     |  82.4 +/- 2.6  |  89.1 +/- 3.7  |
> Ours^      |  88.0 +/- 0.6  |  86.7 +/- 1.3  |
> ---------------------------------------------
> Teacher    |  91.1 +/- 0.4  |  88.9 +/- 1.4  |
> *: Main network parameters are pre-trained with all 50+5 meta-training classes instead of 5 classes.
> ^: Meta-learner parameters are trained with 50 meta-training classes, but the main network parameters are only trained with 5 classes.

---

> > ### Author Response · Authors · 2019-11-14
> > **Response to R1 (part 2)**
> >
> >
> > ------
> > We also thank R1 for providing a detailed list of references. We have already cited most of them and will cite the last two. We carefully checked the reference R1 provided, in particular,
> >
> > Ebrahimi, Sayna, et al. "Uncertainty-guided Continual Learning with Bayesian Neural Networks." has not been published in a conference venue, and it may be too early to compare with it.
> >
> > Lopez-Paz, David, and Marc'Aurelio Ranzato. "Gradient episodic memory for continual learning." and Shin, Hanul, et al. "Continual learning with deep generative replay." and Nguyen, Cuong V., et al. "Variational continual learning” and Aljundi, Rahaf, et al. "Online continual learning with no task boundaries." We argue that these paper have a different setting compared to ours since they require a buffer whereas our method has no storage of past data/gradient. Having past data storage can usually improve the performance and our method can potentially also get a boost. Having a data buffer can also cost a lot of memory storage depending on input/weight dimension. Therefore, we argue it won’t be a fair setting to compare with methods with data buffers.
> >
> > Zenke, Friedemann, Ben Poole, and Surya Ganguli. "Continual learning through synaptic intelligence." has very similar performance to EWC in their paper. We have cited this work already.
> >
> > Serrà, J., Surís, D., Miron, M. & Karatzoglou, A.. (2018). Overcoming Catastrophic Forgetting with Hard Attention to the Task and other pruning based papers. Thanks for pointing out. We have cited and will compare to them in the future. One thing to note is that, in these works, the model needs to know which task ID it is currently dealing with, and thus can turn on the pruning procedure for the next session. This can potentially be a limitation for dynamic incoming tasks.
> >
> > ------
> > References:
> > Chelsea Finn, Pieter Abbeel, and Sergey Levine. "Model-agnostic meta-learning for fast adaptation of deep networks." In ICML 2017.
> > Khurram Javed, Martha White. “Meta-Learning Representations for Continual Learning.” In NeurIPS 2019 (to appear).

---

### Author Response · Authors · 2019-11-14
**General response**

We thank all reviewers for their time and valuable comments. Here is a summary of the main points.

1. We added transferability experiment to Tiny ImageNet as Task B (see R2, R3 response)

2. We added more baselines that use meta training data (see R1, R3 response)

3. We updated Figure 1 to illustrate the significance of doing measurement of forgetting on the representation level. The change in accuracy is less drastic and more stable (with less variance across runs). (see R3 response)

Please see individual responses below for details of the aforementioned items, as well as other minor points.

---

### Decision · Program_Chairs · 2019-12-19

**Decision:**

Reject

**Comment:**

The paper addresses the setting of continual learning. Instead of focusing on catastrophic forgetting measured in terms of the output performance of the previous tasks, the authors tackle forgetting that happens at the level of the feature representation via a meta-learning approach. As rightly acknowledged by R2, from a meta-learning perspective the work is quite interesting and demonstrates a number of promising results.
However the reviewers have raised several important concerns that placed this work below the acceptance bar:
 (1) the current manuscript lacks convincing empirical evaluations that clearly show the benefits of the proposed approach over SOTA continual learning methods; specifically the generalization of the proposed strategy to more than two sequential tasks is essential; also see R1’s detailed suggestions that would strengthen the contributions of this approach in light of continual learning;
(2) training a meta-learner to predict the weight updates with supervision from a multi-task teacher network as an oracle, albeit nicely motivated, is unrealistic in the continual learning setting -- see R1’s detailed comments on this issue.
(3) R2 and R3 expressed concerns regarding i) stronger baselines that are tuned to take advantage of the meta-learning data and ii) transferability to the different new tasks, i.e. dissimilarity of the meta-train and meta-test settings. Pleased to report that the authors showed and discussed in their response some initial qualitative results regarding these issues. An analysis on the performance of the proposed method when the meta-training and testing datasets are made progressively dissimilar would strengthen the evaluation the proposed meta-learning approach.
There is a reviewer disagreement on this paper. AC can confirm that all three reviewers have read the rebuttal and have contributed to a long discussion. Among the aforementioned concerns, (3) did not have a decisive impact on the decision, but would be helpful to address in a subsequent revision. However, (1) and (2) make it very difficult to assess the benefits of the proposed approach, and were viewed by AC as critical issues. AC suggests, that in its current state the manuscript is not ready for a publication and needs a major revision before submitting for another round of reviews. We hope the reviews are useful for improving and revising the paper.